# OBJECTIVE EVALUATION OF DEEP VISUAL INTERPRETATIONS ON TIME SERIES DATA

## ABSTRACT

The correct interpretation and understanding of deep learning models is essential in many applications. (Explanatory) visual interpretation approaches for image and natural language processing allow domain experts to validate and understand almost any deep learning model. However, they fall short when generalizing to arbitrary time series data that is less intuitive and more diverse. Whether a visualization explains the true reasoning or captures the real features is more difficult to judge. Hence, instead of blind trust we need an objective evaluation to obtain reliable quality metrics. This paper proposes a framework of six orthogonal quality metrics for gradient- or perturbation-based post-hoc visual interpretation methods designed for time series classification and segmentation tasks. This comprehensive set is either based on "human perception" or on "functional properties". An extensive experimental study includes commonly used neural network architectures for time series and nine visual interpretation methods. We evaluate the visual interpretation methods with diverse datasets from the UCR repository and another complex real-world dataset. We show that none of the methods consistently outperforms any of the others on all metrics while some of them are ahead in either functional or human-based metrics. Our results allow experts to make an informed choice of suitable visualization techniques for the model and task at hand.

## 1 INTRODUCTION

Due to its high performance on complex multi-modal data, deep learning (DL) becomes increasingly popular in many real-world applications that process time series data (Fawaz et al., 2019b). While we fundamentally rely on their classification accuracy in many safety-relevant applications (Berkenkamp et al., 2017) they remain difficult to interpret. Typical applications are the monitoring of industrial processes (Löffler et al., 2021), the support of health care and sports (Dorschky et al., 2020), or safety in autonomous driving (Schmidt et al., 2021). The need for improved model understanding (Carvalho et al., 2019), along with regulatory guidelines (Goodman & Flaxman, 2017), led to a myriad of new approaches to the visual interpretation problem (Zhang & Zhu, 2018).

Post-hoc visual interpretation allows a domain expert to validate and understand how (almost) arbitrary deep learning models operate. Their central idea lies in highlighting features on the input that are "relevant" for the prediction of a learned model (Adebayo et al., 2018). Many of these techniques do not require a modification of the original model (Simonyan et al., 2014; Ribeiro et al., 2016) and are compatible with different architectures, which makes them useful as a general-purpose validation tool for neural networks across different tasks (Arrieta et al., 2020).

However, while visual interpretation yields intuitive and correct explanations for images (Samek et al., 2021), the application of these methods on time series data is still an unsolved problem (Rojat et al., 2021). Time series inherently are more diverse (Rojat et al., 2021) (because they may originate from a variety of different sensors and processes), and often do not allow an obvious patch- or texture-based localization of critical features for human observers. This makes the application and the evaluation of visual interpretability methods difficult. A domain expert cannot easily judge if explanations are correct in (i) explaining the reason for the decision process in the DL model and in (ii) capturing the real features in the dataset that lead to a correct classification.

Hence, it is important not to blindly apply different visualization methods. This requires quality metrics that evaluate visual interpretations and that enable an expert to select a suitable visualization for

a given model and task at hand. However, both state-of-the-art visualization techniques and metrics that evaluate visual interpretations (e.g. Pixel Flipping (Samek et al., 2017), Sanity Check (Adebayo et al., 2018), and sensitivity checks (Rebuffi et al., 2020)) have so far only been examined on images (Rojat et al., 2021) or on NLP tasks (Arras et al., 2017). This lack of objective and subjective evaluation seriously limits the application and utility of them for time series.

In this paper, we propose to evaluate visual interpretations for time series combining six orthogonal metrics: "sanity" (Adebayo et al., 2018), "faithfulness" (Alvarez Melis & Jaakkola, 2018), "sensitivity" (Rebuffi et al., 2020), "robustness" (Yeh et al., 2019), "stability" (Fel & Vigouroux, 2020; Li et al., 2021), and a novel metric based on human preferences: "localization". These metrics rate and validate distinct qualities of saliency. Our metrics are both based on the functional perspective, i.e., based on the model-specific operation, and on how well they represent annotators' semantics.

In an extensive evaluation, we train four different architectures on two different types of tasks: U-Time model (Perslev et al., 2019) and bidirectional Long Short-Term Memory (bi-LSTM) (Schuster & Paliwal, 1997) on segmentation tasks, and Fully Convolutional Network (FCN) (Long et al., 2015) and Temporal Convolutional Network (TCN) (Bai et al., 2018) on classification tasks. We use diverse datasets from the UCR repository (Dau et al., 2018) (GunPointAgeSpan, FordA, FordB, ElectricDevices, MelbournePedestrian, NATOPS) and for segmentation the more complex real-world tool tracking dataset (Löffler et al., 2021). The experiments show the necessity of all categories to create an objective rating for methods, models and tasks, and allow domain experts to understand, rate, and validate saliency for time series in safety-critical applications.

The rest of the paper is structured as follows. Section 2 discusses background and related work. Section 3 introduces extended and novel metrics for both the classification and segmentation task. Section 4 discusses the experimental results and Section 5 proposes recommendations.

## 2 BACKGROUND AND RELATED WORK

Interpretation methods for DL models can be divided into ante-hoc methods, i.e., methods that are inherently part of the model, and post-hoc methods, i.e., methods that provide the interpretation after training (Rojat et al., 2021). We focus on post-hoc methods and divide them into gradient-based and perturbation-based methods (Li et al., 2021; Warnecke et al., 2020; Ismail et al., 2020).

*Gradient-based* methods compute the relevance for all input features by passing gradients backwards through the neural network (Ancona et al., 2018). Gradient (Simonyan et al., 2014) computes a class $c$'s saliency map $M^c$ using the derivative of the class score $P^c$ of the model with respect to the input sample $x$, as $M^c(x) = \frac{\partial P^c}{\partial x}$. The gradient indicates the importance of points in the input sequence for predictions. The advantage of gradient-based methods lies in their computational efficiency, as they use only a small number of backward passes to compute $M^c(x)$.

*Perturbation-based* methods perturb known input samples and measure the effects of specific perturbations on the predicted class via a forward pass through the network. For instance, Local Interpretable Model-Agnostic Explanations (LIME) (Ribeiro et al., 2016) fits a local surrogate model (e.g., a linear regression) as an explanation, and calculates relevance based on this surrogate. Perturbation-based methods are computationally expensive as they require multiple forward passes per sample. However, they do not need gradient information and work with black-box models.

### 2.1 METRICS FOR SALIENCY ON TIME SERIES

Most interpretation methods were originally designed for image or text data. Understanding and comparing visual interpretation is intuitive on image data, compared to more abstract time series. Furthermore, the diversity of interpretation methods complicates an objective choice for the model and task (Rojat et al., 2021). For example, when Wang et al. (2017) and Fawaz et al. (2019b) apply Class Activation Maps (CAM) (Zhou et al., 2016) on well-known UCR datasets (Dau et al., 2018), they notice a qualitative difference of CAM interpretations between network architectures. Similarly, other work relies on domain experts that perform a (costly) qualitative evaluation (Strodthoff & Strodthoff, 2019; Fawaz et al., 2019a; Oviedo et al., 2019). Hence, there is an increasing need for objective evaluation metrics to make the interpretations measurable and comparable.

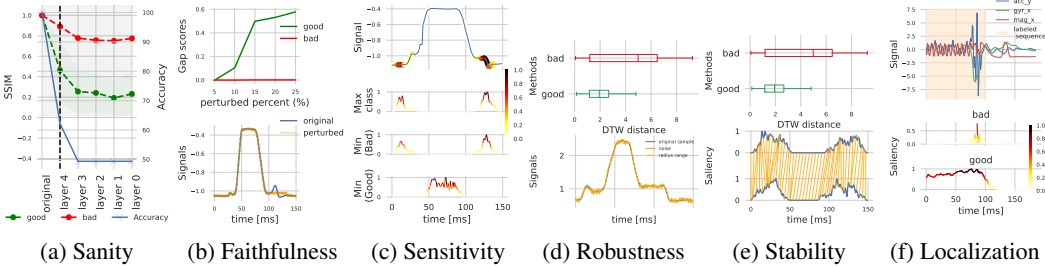

| (a) Sanity | (b) Faithfulness | (c) Sensitivity | (d) Robustness | (e) Stability | (f) Localization |

Figure 1: We show examples for a good (green) and a bad (red) score for each metric. (a) sane saliency depends on network parameters, tested by randomizing weights and biases. (b) faithful saliency correlates with predictive accuracy, tested by perturbing the input sequence. (c) sensitive saliency of the predicted class in one sample is different from others. (d) for robust saliency small changes to input data cause only small effects. (e) stable saliency of all class samples has a low variance and standard deviation. (f) saliency should be localized on the predicted segment.

**Metrics.** There exists a large variety of different metrics. First, the stability of saliency maps can be evaluated with respect to the *sensitivity* of the majority class (Yeh et al., 2019). This metric evaluates the change of interpretation when the input samples are attacked by adversarial noise. It is a special form of perturbation, that was so far only applied to image data. Second, Rebuffi et al. (2020) measure class sensitivity by comparing saliency maps of a min- and max-class. Third, Cho et al. (2020) perturb the unimportant input samples and preserve only the important inputs. If the predictions are stable compared to the unperturbed input, the interpretation is robust. Similarly, Ates et al. (2021) evaluate the *local robustness* (i.e., similar samples should lead to similar interpretations) of the interpretation by measuring the ratio of change of the interpretation compared to the amount of input perturbation. Finally, *faithfulness* is a perturbation-based metric that aims to measure the relationship between saliency and predictive features (Alvarez Melis & Jaakkola, 2018). Schlegel et al. (2019) focus on continuous univariate time series. Sub-sequences with high relevance values are perturbed either by swapping their samples or replacing them with a mean value. Similarly, Ismail et al. (2020) propose input perturbations to evaluate models on synthetic time series. As they perturb inputs point-wise instead of sub-sequences their metric does not consider time dependency between consecutive points. Our faithfulness metric perturbs sub-sequences of high saliency (Schlegel et al., 2019). We adapt each type of metric into our set to provide a visual interpretation of time series.

**Categories of metrics.** Metrics can be divided into different categories, depending on the question they answer. Doshi-Velez & Kim (2017) propose a distinction between *human-grounded* and *functional* metrics. The former involve human perception and intuition with the goal of generating qualitative, intuitive visualizations, e.g., bounding boxes for image object detection for testing the localization of saliency maps (Jianming et al., 2016), or questionnaires to indicate testers' opinions on the quality of explanations (Li et al., 2021). Functional metrics provide statistical measures, e.g., to aggregate performance metrics automatically (Rojat et al., 2021), and make use of proxy tasks to generate a quantitative evaluation. Another categorization (Li et al., 2021) identifies multiple broad categories for image data, that we may transfer to time series, i.e., the faithfulness of salient features, the class sensitivity of an explanation, and the stability of explanations given noisy inputs.

This paper proposes a framework of metrics with orthogonal categories, specifically to time series. We adapt and extend metrics to (multivariate) time series and propose an intra-class stability metric and a concept of relevance localization: we build on the pointing game (Jianming et al., 2016) and combine it with the precision and recall for time series framework (Tatbul et al., 2018). To our knowledge we are the first to apply and evaluate sanity check (Adebayo et al., 2018) on time series.

## 3    SCORING CATEGORIES

We propose a set of six distinct categories (sanity, faithfulness, sensitivity, robustness, stability, and localization) to assess visual interpretation methods and to determine their performance and trustworthiness in classification or segmentation tasks on time series. For each of them we propose a metric that enables a comparative evaluation of diverse types of visual interpretation methods.

**Why do we need six scoring categories?** It seems tempting to rely on a single metric or on a single aggregated score across multiple metrics to assess the quality of a visual interpretation. However, we show that the six presented categories are inherently orthogonal (see Fig. 2) and capture distinct qualities. Interpretations depend on model parameters (sanity check (Adebayo et al., 2018)), predictive features (faithfulness (Alvarez Melis & Jaakkola, 2018)), coherence of class predictions (intra-class stability (Fel & Vigouroux, 2020)), robustness against noise (max sensitivity with adversarial noise (Yeh et al., 2019)), specific sequences (or even points) in a time series like by Tatbul et al. (2018) (novel localization metric), and the relevance map's specificity (inter-class sensitivity (Rebuffi et al., 2020)). It is important to assess if a given interpretation accurately captures these dependencies.

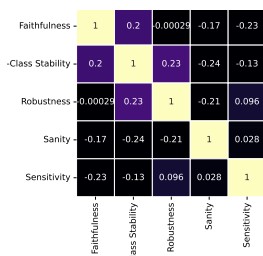

Figure 2: Pairwise Pearson correlations of scores (also plotted in Fig. 7b). Every metric is independent of all others.

We give a brief overview of our frameworks' functional (Fig. 1a-e) and human-grounded metrics (Fig. 1f) that we evaluate with time series data. (a) *sane saliency* depends on network parameters and is structurally different after randomizing the network's weights $\rho_i$ for layers $[1, 2, 3]$ (Adebayo et al., 2018). (b) *faithful saliency* correlates with predictive accuracy and perturbing a percentage of the input sequence with high saliency decreases accuracy (Alvarez Melis & Jaakkola, 2018). (c) saliency is *sensitive* when the predicted (max) class in one sample is sufficiently different from any other (min) class (Rebuffi et al., 2020). (d) saliency is *robust*, if small changes to the input cause only small changes to the saliency (Yeh et al., 2019). (e) saliency is *stable* if it has a low variance and standard deviation for all samples of a class, with respect to a suitable distance metric (Fel & Vigouroux, 2020). (f) saliency should be *localized* on the predicted classes segment ($t_0$ to $t_1$). We will define those categories on time series after introducing a unified notation.

**Notation.** We follow the notation adapted from Fawaz et al. (2019b): A multivariate time series is defined by $X = [X^1, ..., X^H]$, where $H$ is the number of input channels, $X^i = (x_1^i, ..., x_T^i) \in \mathbb{R}^T$ is an ordered set of real values, and $T$ denotes the number of time steps. For $H$ equal to 1, we consider a *univariate* time series, otherwise we consider a *multivariate* time series. Time series often include complex temporal dependencies, i.e., distinct points are not independent. Time series classification defines a mapping $X \rightarrow y$ that minimizes the error on a dataset $D = \{(X_1, y_1), ..., (X_N, y_N)\}$, where $N$ is the number of data samples, $X \in D$ is a time series, $y_i \in \mathbb{R}^C$ denotes the one-hot vector of a class label the input belongs to, and $C$ is the number of classes. In time series segmentation, we search $X \rightarrow Y$ that maps an input sample to a dense classification $Y = [y_1, ..., y_T] \in \mathbb{R}^{C \times T}$ (Perslev et al., 2019), i.e., a class label is predicted for each time step. Post-hoc visual interpretation methods compute a relevance map $M_m^c \in \mathbb{R}^{H \times T}$, $M_m^c(X) = [R^1, ..., R^H]$, where $R^i = (r_1^i, ..., r_T^i)$, representing the importance of each input feature with respect to the class $c$ and a model $m$, for each time step. We use $M$ as a function to produce the saliency map. For clarity, we will omit the dependency on $m$, i.e., $M_m^c \equiv M^c$, if it is not explicitly required. An evaluation metric for visual interpretation methods defines a score $\mathcal{S}_{\text{metric}}(\cdot)$ that rates the quality of the relevance map $M$ at a sample $X$ given a model $m$ and optional parameters. We provide a unified view, so that for all scores, a higher score corresponds to a better visualization according to the perspective.

### 3.1 SANITY

Intuitively, if the weights and biases of trained networks models were re-initialized with random values, the networks predictions and generated saliency maps should also be different from the original maps. However, this is not always the case. Despite a drop in model accuracy, saliency may remain stable. Hence, we test sanity using a variant of sanity check (Adebayo et al., 2018), that performs a *layer-wise cascading randomization* of the network's weights and biases, starting from the output to the input. In contrast to independent randomization, the cascading approach results in a mostly continuous performance degradation of predictions, see Fig. 1a for an illustration. Network accuracy should increasingly resemble random guessing. Following Adebayo et al. (2018), we compare saliency using the structural similarity index measure (SSIM) (Wang et al., 2004), that compares the distribution of sub-sequences of $M^c$. We define the sanity score as

$$S_{\text{Sanity}}(M, D, m) = -\frac{1}{N} \cdot \sum_{X \in D} \sum_{i=1}^L \frac{\text{SSIM}(M_m^c(X), M_{m_i}^c(X))}{L} \tag{1}$$

where $|D| = N$, $i$ enumerates the $L$ layers of $m$, whose parameters are randomized. $M_{m_i}^c(x)$ is the saliency map after randomizing layer $i$ of $m$. We average the SSIM over $L$ layers and compute the average over all samples in $D$.

### 3.2 FAITHFULNESS

A relevance measure is faithful, if input features with a high relevance (w.r.t. $M^c$) have a high influence on the model prediction, see Fig. 1b for an example. Alvarez Melis & Jaakkola (2018) propose a perturbation-based metric that evaluates the faithfulness of predictions. The metric measures the correlation between input (features) with high saliency on the one hand and predictive accuracy on the other hand. We choose a sub-sequence of the input to perturb by ranking each time step of $X_T \in \mathbb{R}^{\mathbb{T}}$ according to the relevance of its saliency map. Next, we perturb connected sub-sequences similar to Schlegel et al. (2019). We extend the metric to multivariate data $X_T^H$: Based on the relevance $r$, we select time points $t \in \{1, ..., T\}$ and features $i \in \{1, ..., H\}$. We select $t$ according to the maximum relevance $r_t^i$ and set $t$ as the middle point of the sub sequence. Therefore, we select and reverse the sequence as $x_{\text{rev}} = (x_{t+\frac{L}{2}}^i, ..., x_t^i, ..., x_{t-\frac{L}{2}}^i)$ to break the temporal correlation. Next, we insert it back into a copy of $X$ that we denote as $X'$. With this perturbation we compute the mean faithfulness score over the whole dataset $D$ as

$$S_{\text{Faithfulness}}(M, D, m) = \frac{1}{N} \cdot \sum_{X \in D} m^c(X) - m^c(X'). \quad (2)$$

where $m^c : R^c \to R$ is the softmax prediction of the target class $c$. $m^c(X) - m^c(X')$ is the gap score between the softmax prediction of the original and the perturbed sample.

### 3.3 INTER-CLASS SENSITIVITY

In multi-class prediction tasks the classifier needs to identify relevant features for each of the classes to make a correct prediction. Hence, the relevance map $M^c$ should identify different salient features for those different classes (Li et al., 2021), see Fig. 1c. If a method is not sensitive to the class, the saliency could be misleading, especially if the classifier fails to learn correct features for some classes. Inter-class sensitivity (Rebuffi et al., 2020) measures class specific sensitivity of the generated relevance map with respect to the most ($c_{\max}$) and least ($c_{\min}$) likely class according to the model. We compute the mean inter-class sensitivity score as

$$S_{\text{Inter-Class Sensitivity}}(M, D) = -\frac{1}{N} \cdot \sum_{X \in D} \text{sim}(M^{c_{\max}}(X), M^{c_{\min}}(X)). \quad (3)$$

We compute similarity of two saliency maps as $\text{sim}(M^{c_{\max}}(x), M^{c_{\min}}(x))$ where $\text{sim}(\cdot, \cdot)$ is a similarity function (e.g. a cosine similarity) that is easy to interpret via its geometric interpretation. It is defined as the angle between two non-zero vectors that measures the similarity between their inner product space (Han et al., 2012). Similarity of $M$ in binary classification would result in a negative cosine similarity, meaning nearly inverted saliency maps for max- and min-classes.

### 3.4 ROBUSTNESS

We evaluate a method's robustness with respect to adversarial noise (Yeh et al., 2019) via its sensitivity of the most likely class ($c_{\max}$) see Fig. 1d. Intuitively, even with noisy inputs the saliency map of a model should not change significantly (Alvarez-Melis & Jaakkola, 2018) and saliency maps with high sensitivity are less reliable. Yeh et al. (2019) define the sensitivity of the saliency map derived from the gradient as

$$[\nabla_X M^c(X)]_j = \lim_{\epsilon \to 0} \frac{M^c(X + \epsilon e_j) - M^c(X)}{\epsilon} \quad (4)$$

for any $j \in \{1, ..., |H \cdot T|\}$, where $e_j \in \mathbb{R}^{H \times T}$ is the $j$-th coordinate vector and the $j$-th entry is one while the others are zero. We use Monte Carlo sampling of $\epsilon e_j$, where $|\epsilon e_j| < a$ ($a$ is a user-specified radius), to generate different $\hat{X} = X + \epsilon e_j$. We compare $\hat{X}$ with the original $X$ to compute

$$S_{\text{Max Sensitivity}}(M^c, D, a) = -\frac{1}{N} \cdot \sum_{X \in D} \max_{||\hat{X} - X|| < a} ||M^c(\hat{X}) - M^c(X)||. \quad (5)$$

### 3.5 INTRA-CLASS STABILITY

A method is stable if similar input samples from the same class produce similar interpretations (Fel & Vigouroux, 2020). We implement this concept to test the intra-class stability of saliency maps $M^c$ for a given dataset $D$ using distance statistics, see Fig. 1e for an example. To this end, we compute pairwise distances between saliency maps $M^c$ for different samples $X_i, X_j \in D$. Then, we aggregate these distances for all samples from one class:

$$S_{\text{Intra-Class Stability}}(M^c, D) = -\sum_{i \in [0,N]} \sum_{j \in [i+1,N]} \frac{d_{\text{dtw}}(M^c(D_i), M^c(D_j))}{N \cdot (N-1)}. \tag{6}$$

We use Dynamic Time Warping (Vintsyuk, 1968) as the distance function $d_{\text{DTW}}$. The score compares each classes' sample's $M^c$ with all other samples' saliency maps in the dataset, using the distance $d_{\text{DTW}}$. In summary, the mean distance for a class should be small and the variance low.

### 3.6 LOCALIZATION

The temporal location of class-specific features with high relevance in a time series segmentation task should be situated within (or close-by) its labeled segment (temporal sub-sequence), see Fig. 1f. Due to its connection to the annotation task we call it a *human-grounded* metric. Specifically, we propose a novel metric that adapts Pointing Game (Jianming et al., 2016) used for object detection to time series using range-based metrics (Tatbul et al., 2018). We argue that Pointing Game's original hit-and-miss accuracy is inadequate for time-series methods, as Segmentation is point-wise exact. Thus, we replace it with range-based testing, yielding Localization. Furthermore, we observed that some saliency methods can show strong temporal biases towards the beginning, middle or end of a segmented class. We would expect high saliency for features within the predicted segment, and low saliency otherwise, to accurately capture the model's predictive behavior. Hence, Localization measures agreement between annotators, models' predictions and saliency, and can discover issues like temporal biases and imprecise margins.

First, we filter out non-relevant predictions (according to the relevance map) as follows. A prediction at time $t$, denoted with $\hat{Y}(t)$, is relevant if there exists an $i \in \{1, ..., M\}$ such that $r_t^i > \max(|r|) \cdot \theta$. We select the model's predicted class at time $t$ if the prediction is relevant, otherwise we set the class to *none*. The resulting *relevancy-filtered* prediction is denoted as $\hat{Y}'$. Finally, we can compare $\hat{Y}'$ with the ground truth $Y$ and evaluate how well high saliency features lie within the annotated sub-sequences (Tatbul et al., 2018).

We compute the mean localization score for the whole dataset $D$. We compare any of the $N_{\text{sub}}$ existing labeled sub-sequence $Y_{\text{sub}_i} \in R^T$ from $D$ each with its temporally co-located prediction $\hat{Y}'_i$

$$S_{\text{localization}}(Y, \hat{Y}') = \sum_{i=0}^{N_{\text{sub}}} \frac{S_{\text{recall}}(Y_{\text{sub}_i}, \hat{Y}'_i)}{N_{\text{sub}}}. \tag{7}$$

For each pair $(Y_{\text{sub}_i}, \hat{Y}'_i)$ we calculate a range-based recall score based on the point-wise comparison proposed by Tatbul et al. (2018) as

$$S_{\text{recall}}(Y_{\text{sub}}, Y') = \alpha \cdot \text{existence}(Y_{\text{sub}}, Y') + (1 - \alpha) \cdot \text{overlap}(Y_{\text{sub}}, Y'), \tag{8}$$

where $\alpha$ weighs the two reward terms for "existence" and "overlap". Existence is 1 if any time point was correctly predicted within the labeled region, and 0 otherwise. The parameterized "overlap" function determines the finer properties of cardinality, size and position. The cardinality parameter discounts the score if the prediction is an interrupted, fragment range instead of being continuous. The overall size of the overlap of predicted and label ranges depends on a positional bias. It may favor "front", "middle", or "back" overlap. Practically, for some applications an early detection is preferable over a late detection. For further details see Appendix A or Tatbul et al. (2018).

## 4 EXPERIMENTS

Our evaluation compares visual interpretation methods on a set of network architectures and diverse classification datasets from the UCR repository, and on a more complex segmentation dataset. In total, we aggregate the results from 540 experiments. This section is divided into two parts. We first discuss the new localization metric for the segmentation task. Next, we discuss the metrics for the classification task (faithfulness, sensitivity, stability, robustness, sanity).

### 4.1 EXPERIMENTAL SETUP

We evaluate nine visual interpretation methods: Gradient (Simonyan et al., 2014), Integrated Gradient (Sundararajan et al., 2017), SmoothGrad (Smilkov et al., 2017), Guided Backpropagation (Springenberg et al., 2015), GradCAM (Selvaraju et al., 2017), Guided-GradCAM (Selvaraju et al., 2017), Layer-Wise Relevance Propagation (LRP) (Bach et al., 2015), LIME (Ribeiro et al., 2016) and Kernel SHAP (Lundberg & Lee, 2017). See Appendix B.2 for method-specific hyper parameters.

For the segmentation task, we consider two models: U-time (Perslev et al., 2019) (derived for time series from U-Net (Ronneberger et al., 2015)), and a bi-LSTM (Schuster & Paliwal, 1997). We provide details on each architecture in Appendix B.3. We evaluate them on the tool tracking (Löffler et al., 2021) dataset, a complex multivariate, multi-class time series from a 9-D magneto-inertial sensor, and use the electric screwdriver's data. For the classification task, we select two commonly used model architectures, i.e, a Fully Convolutional Network (FCN) (Long et al., 2015) and a Temporal Convolutional Network (TCN) (Bai et al., 2018). We do not focus on simpler architectures such as Multilayer Perceptrons and LSTMs (Hochreiter & Schmidhuber, 1997) due to noisy or vanishing saliency in preliminary studies. Our dataset selection follows related work (Fawaz et al., 2019b; Wang et al., 2017; Schlegel et al., 2019; Ates et al., 2021) that uses large, multivariate, multi-class datasets from diverse domains (GunPointAgeSpan, FordA, FordB, ElectricDevices, MelbournePedestrian and NATOPS from the UCR repository (Dau et al., 2018)).

### 4.2 EVALUATION ON SEGMENTATION TASK

This section presents the key findings from our evaluation when applying the localization metric on saliency maps generated from U-time (results for bi-LSTM are in Appendix C.1), trained for a segmentation task on the tool tracking dataset. Fig. 3a shows the localization metrics' positional biases "classic", "front", "middle" and "back" for each visualization method. Fig. 3b shows an exemplary saliency map for each method.

GradCAM clearly outperforms the other methods on the localization metric. It's saliency maps are smoother, with high relevance located on the annotated segments. This may result from its coarser relevance, as its gradients do not flow to the input but stop at the last convolution layer. In contrast, LRP and Integrated Gradient perform worst. As Fig. 3b shows, the saliency maps highlight fewer relevant input features for the segmentation task, but instead features within the labeled sub-sequence that may be important for a classification. Saliency maps are located better in the "middle" of the annotated segments, compared to "front"

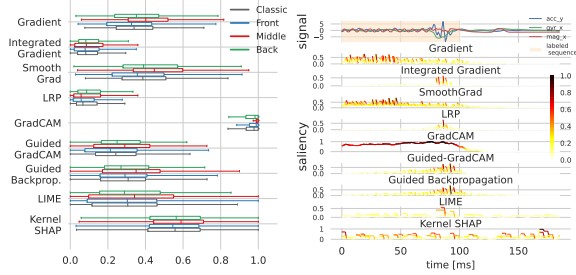

(a) U-time localization.  (b) U-time examples.

Figure 3: Results for the localization metric on the tool tracking dataset and with all methods on U-time. We show detailed values for the localization metric in (a) and examples of saliency maps in (b).

and "back", see Fig. 3a. This is also visible in the example in Fig. 3b where the relevance values start decreasing below the threshold of 0.5 already before the end of the labeled segment at 100 ms. This points towards a biased labeling process with longer sub-sequences assigned to the classes than necessary, and can help identifying faulty labels.

Furthermore, the high variance of most results (except for GradCAM) reveals that saliency maps for time series segmentation are noisy. This is especially the case for Kernel SHAP, see Fig. 3b, where relevance is assigned seemingly at random. In summary, GradCAM performs best with respect to the human-grounded metric of localization, compared to all other methods in this study. It exhibits low variance in the results and focus on the "middle" segment, while other methods hardly align with annotated relevant segments in the input features or their saliency maps are noisy.

Conversely, no method works reliably on the bi-LSTM. We identify a lack of useful visualization methods for LSTMs, as CAM variants (GradCAM, Guided GradCAM, and Guided Backprop) are not applicable, see Fig. 6a and 6b in Appendix C.1 for detailed results.

## 4.3 EVALUATION ON CLASSIFICATION TASK

In this section, we discuss the five visual interpretation quality metrics for the classification task. Fig. 4a summarizes the scores for each metric over model architectures (FCN, TCN) and datasets.

**Sanity.** Across all datasets, we find that Gradient, Integrated Gradients, LIME, and Kernel-SHAP consistently achieve high sanity scores. In contrast, LRP, Guided GradCAM and Guided Backpropagation often assign similar saliency even after randomizing network parameters. This replicates sanity results by Adebayo et al. (2018). Low sanity methods highlight features in the input that may also be randomly extracted but do not necessarily predict classes.

**Faithfulness.** The scores differ widely across datasets. While Gradient, Integrated Gradients, LRP and Guided Backprop have an above-average faithfulness, Guided Gradcam and GradCAM are unable to identify critical features for a prediction. Especially scores for FordA, FordB and NATOPS are generally very low across all visualization methods. Perturbating connected sub-sequences instead of random samples may be less suitable for these datasets.

**Sensitivity.** GradCAM shows the highest sensitivity for classes - the saliency maps for the most and least likely classes differ largely. On the other hand, Integrated Gradient and Guided Backpropagation produce similar saliency maps, independent of a sample's class. This holds across all datasets.

**Robustness.** As found by Smilkov et al. (2017) and us, Gradient is less robust to (adversarial) noisy. LIME and, interestingly, SmoothGrads also have a low average robustness. The additive noise of SmoothGrads may compound and lead to diverging saliency maps. The metric's weakness is computational cost – {TCN,ElectricDevices} timed out after 12 days.

**Intra-class Stability.** This metric is highly dependent on the dataset, e.g., centered samples as in GunPointAgeSpan. However, when accounting for variance introduced by the dataset (see Fig. 7b in Appendix C.2), SmoothGrads, LIME, and Kernel-SHAP show unstable saliency maps for the same classes. Guided GradCAM is able to produce highly stable saliency maps across all datasets.

**Dataset Influence.** First, datasets heavily impact scores, see the outliers in Fig. 4a. Plotting the categories over each dataset for all {model architectures, methods} (Fig. 4b) emphasizes this finding. The choice of visual interpretation method depends, for the most part, on the task. Still, after normalizing for dataset bias, the relative scores do not diverge significantly but confirm the initial scores. We report the relative scores in Fig. 7b in Appendix C.2. Second, the model architecture (FCN or TCN) matters little to the scores, see Fig. 7c in Appendix C.2. The scores for FCN and TCN diverge slightly more for faithfulness. This may be due to the higher capacity of the TCN, which learns more features of lower relevance, so that accuracy is not as affected by perturbations as for the smaller FCN. A larger perturbation percentage may resolve this issue.

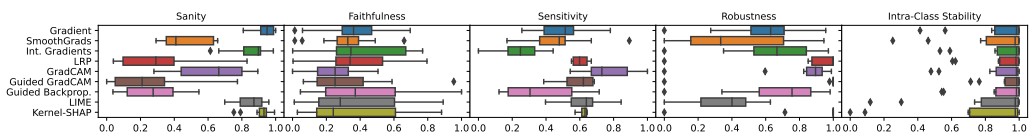

(a) Scores for all nine methods aggregated over {model architectures, datasets}.

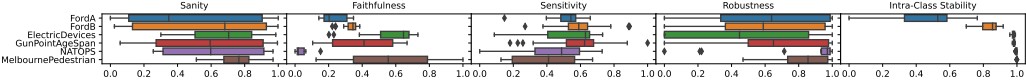

(b) Datasets' impact on scores, aggregated over {model architectures, methods}.

Figure 4: (a): shows aggregated results for TCN and FCN over all classification datasets without normalization ( as we found that model architectures generally lead to similar scores); (b): shows the influence of datasets on the scores, when we aggregate for each dataset over {model architectures, methods}. Similarity metrics for saliency maps yield different results in different domains.

## 5 RECOMMENDATIONS

The quality of visualizations differs wildly between methods. No method passes the tests for all categories on all six datasets. This emphasizes the need to evaluate all metrics for every visual interpretation. We recommend to use a summary as in Fig. 5 to judge visualizations on every category, and propose the following guidelines for relative ranking. The absolute scores may be understood in comparison with a random baseline, similar to shuffled AUC (Borji et al., 2013).

**First: Ensure Faithfulness and Sanity.** The general purpose of interpretability methods is to provide insights into model behavior. We propose to use the Faithfulness and Sanity scores to ascertain that a saliency map represents the model behavior. Faithfulness ensures that the saliency matches the model's predictive features. Sanity checks confirm that saliency maps are sensitive to model parameters. This is important to avoid finding highly salient features, like edges in images (Adebayo et al., 2018), while being insensitive to model parameters. We avoid saliency maps with low scores in either metric. See Fig. 5 for an example: Gradient, IG and LIME achieve the three highest scores. Random saliency maps are dissimilar, hence the high Sanity score.

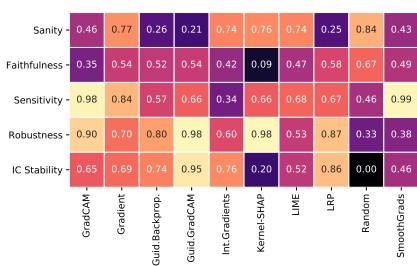

Figure 5: Scores as a heatmap for TCN on FordB. Columns with high relative scores (bright squares) indicate good visualizations. Dark squares show the shortcomings.

However, its Faithfulness hints at low reliability of the other methods. Guided Backprop, Guided GradCAM and LRP fail the Sanity Check, while Kernel-SHAP performs poorly in Faithfulness.

**Second: Check Sensitivity and Robustness.** Once Faithfulness and Sanity are established, we propose to look at Sensitivity and Robustness of the generated saliency maps. A low Inter-Class Sensitivity can indicate that the saliency maps only focus on the predicted class and underestimate the importance of features that do not belong to this class. A low Robustness score suggests that the visualization method is susceptible to adversarial examples and small perturbations in the input. Given that the model predictions are robust (Zhang et al., 2019), and the saliency is faithful, a low Robustness implies that the saliency method cannot be trusted, and may even be manipulated (Dombrowski et al., 2019). For non-robust models we recommend Faithfulness. According to Fig. 5, we keep Gradient and LIME, but disregard IG, due to its lower Sensitivity.

**Detailed Analysis: Analyze Intra-class Stability and Localization.** Intra-class stability measures how much saliency maps for one class agree between different samples. The localization metric, like other Pointing Game-like metrics, measures semantic precision based on human annotations. These human-grounded metrics allow an expert to filter visualizations that are more intuitive and understandable. Note that it is crucial to ensure the Faithfulness of a method before relying on this metric. If the method is not faithful, stable and localized visualizations are visually pleasing, but do not reflect model behavior. This can hide issues like spurious correlations (Arjovsky et al., 2019) or the shortcut learning problem (Geirhos et al., 2020) behind a good score.

## 6 CONCLUSION

This paper proposes a framework of six orthogonal metrics for the objective evaluation of visual interpretation methods for time series classification or segmentation. All these metrics should be evaluated for every visual interpretation to prevent a reliance on interesting, but spurious results. We also show that each perspective emphasizes different strengths and weaknesses of visualization methods. We show empirically for six datasets and different model architectures that none of nine state-of-the-art visual interpretability methods passes all tests. We propose to use our framework to guide the selection of visual interpretations and to understand their weaknesses. Further, we found that datasets highly influence the quality of visualizations. Our extended and novel metrics provide important information, like the localization metric for time series. Interestingly, on the segmentation task only one visual interpretation method passes our novel evaluation metric. None of the saliency maps for LSTMs produce satisfactory scores. Future work combines range-based and functional metrics, or optimizes models specifically to achieve higher interpretability scores.

REPRODUCIBILITY

This paper includes supplemental materials to improve reproducibility. Most importantly, we publish the code for model training and dataset loaders, for generating visual interpretations and of all evaluation metrics. Furthermore, all datasets are publicly available.

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

## A  LOCALIZATION METRIC POSITIONAL BIASES

This section explains the "existence" and "overlap" terms of our localization metric. Note again the recall score as

$$S_{\text{recall}}(Y_{\text{sub}}, Y') = \alpha \cdot \text{existence}(Y_{\text{sub}}, Y') + (1 - \alpha) \cdot \text{overlap}(Y_{\text{sub}}, Y'), \qquad (9)$$

The "existence" and "overlap" terms are defined by Tatbul et al. (2018) as follows. First, existence is defined as a correct prediction of one sample point at index $j$ of the correct class within the labeled region:

$$\text{existence}(Y_{\text{sub}}, Y') = \begin{cases} 1, & \text{if } \sum_{j=1}^{|Y'|} |Y_{\text{sub}} \cap Y'_j| \geq 1 \\ 0, & \text{otherwise} \end{cases} \qquad (10)$$

The overlap determines the finer properties cardinality $\gamma()$, size $\omega()$ and position $\delta()$.

$$\text{overlap}(Y_{\text{sub}}, Y') = \text{cardinality}(Y_{\text{sub}}, Y') \cdot \sum_{j=1}^{T} \omega(Y_{\text{sub}}, Y_{\text{sub}} \cap Y'_j, \delta) \qquad (11)$$

$$\text{cardinality}(Y_{\text{sub}}, Y') = \begin{cases} 1, & \text{if } Y_{\text{sub}} \text{ overlaps with at most one } Y'_j \in Y' \\ \gamma(Y_{\text{sub}}, Y'), & \text{otherwise} \end{cases} \qquad (12)$$

The term $\alpha$ ($0 \leq \alpha \leq 1$) weights existence and the following qualitative measures of overlap. The three user-defined functions return values of $0 \leq \gamma() \leq 1$, $0 \leq \omega() \leq 1$ and $\delta() \geq 1$. The

cardinality $\gamma()$ function weighs the prediction of a continuous range $Y'$ covering the whole labeled range $Y_{\text{sub}}$ versus interrupted ranges in a fragment manner. The overall size $\omega()$ of the agreement of predicted and label ranges depends on the positional bias $\delta()$ and can favor "front", "middle", or "back". Practically, for some applications an early detection is preferable to a late detection. For the four functions, we selected $\alpha$ is equal to 0, $\gamma$ is equal to 1. $\omega$ depends on the position bias $\delta$ and for $\delta$ we have three variants, "front", "middle" or "back", proposed by Tatbul et al. (2018).

## B EXPERIMENTAL SETUP

We describe each visual interpretation method shortly in Sec. B.1, then the hyper parameters for each visual interpretation method in Sec. B.2 and for the optimizer and networks in Sec. B.3.

### B.1 CATEGORIZATION OF METHODS

This Section provides a short introduction to the methods that we use in our experiments.

**Gradient-based methods.** Gradient (Simonyan et al., 2014) computes class $c$'s saliency map $M^c$ using the derivative of the class score $P^c$ of model with respect to the input sample $x$, as $M^c(x) = \frac{\partial P^c}{\partial x}$. However, Gradient suffers from the *saturation problem* (a feature may have global importance, but its local derivative is small (Smilkov et al., 2017)), local sensitivity, and noisy saliency maps (due to (sharp) local variations in the gradients (Smilkov et al., 2017)). Follow-up work smooths the gradients to reduce noise (Smilkov et al., 2017), applies special propagation rules instead of propagating a gradient (Arras et al., 2017), or propagates only up to a specific intermediate layer (Selvaraju et al., 2017).

**Perturbation-based methods.** Local Interpretable Model-Agnostic Explanations (LIME) (Ribeiro et al., 2016) fits a local surrogate model (e.g., a linear regression) as an explanation and uses this surrogate to calculate relevance. Kernel SHAP (Lundberg & Lee, 2017) builds on LIME, but calculates Shapley values that measure the contribution of individual features to the input more accurately.

### B.2 VISUALIZATION METHODS HYPER PARAMETERS

We set the hyper parameters for each visual interpretation method according to their recommendations from literature. We provide the reasoning behind the selected parameters in the following paragraphs.

Gradient computes the derivative of the target class score with respect to the input sample and returns the saliency map of the input sample at the end. Integrated Gradient uses the linear path method to compute gradient along the path from a baseline $x'$. We use a zero vector for the baseline. As suggested by Smilkov et al. (2017), the number of steps for the path should be selected between 20 and 300. Hence, we use steps $N = 60$, meaning that it takes 60 steps from baseline $x'$ to the original input sample $x$, according to $x = x' + \frac{(x-x')}{N} \cdot n$, $n$ is the current step.

SmoothGrad also computes the derivative of the target class score with respect to the input sample. However, it adds Gaussian noise $\mathcal{N}(0, \sigma^2)$ to the input sample multiple times and computes the gradients from the perturbed samples $(x + \mathcal{N}(0, \sigma^2))$. The number of iteration of adding noise is chosen $N = 60$ and the standard deviation of Gaussian noise is chosen $\sigma = 0.2$.

For LRP, we select the $\epsilon$-propagation rule for every DL model with $\epsilon = 1e - 9$. Due to the residual block in TCN model the propagated relevances should therefore be added together ($R = R_1 + R_2$).

GradCAM and Guided-GradCAM are designed for models with convolutional layers, i.e. FCN, TCN and U-time. GradCAM focuses on the last convolutional layer, which produces feature maps, whose shapes usually are smaller than the shape of input samples. Therefore, we use interpolation to up-sample the saliency maps from the last convolution layer to match their shape to the shape of the input samples, which allows us to visualize them in the input space. Because of ReLU functions, GradCAM returns only positive relevances.

For LIME, we use the cosine distance function as the kernel function with width $w = 5.0$ to weight the perturbed samples, and perform 1000 iterations. For the perturbation, we consider that neighbors along time dimension should have similar relevance to reduce the computational time, so we set the number of features along time dimension 50 for GunPointAgeSpan. This means that saliency maps for samples with length 150 in GunPointAgeSpan have same relevance for every 3 neighbors and saliency maps for dense labeling samples in tool tracking have same relevance for every 4 neighbors.

For Kernel SHAP, we use 1000 iterations and set the number of features along time dimension to 50. Furthermore, the sampling of feature perturbation in Kernel SHAP is based on the distribution $p(f) = \frac{(F-1)}{(f \cdot (F-f))}$, where $f$ is the number of selected features and $F$ is the total number of features in interpretation space.

### B.3 NETWORK ARCHITECTURES

We use the Adam optimizer with a learning rate of $0.002$. We train every dataset for $600$ epochs (with early stopping after $80$ epochs). We use a Cross Entropy loss for time series classification and a Generalized Dice loss with Cross Entropy function for time series segmentation.

For the time series segmentation task on the tool tracking dataset, we report an accuracy of $83\%$ for U-time and $85\%$ for the bi-LSTM. Table 1 shows the classification results for FCN and TCN with all classification datasets.

| | model architectures | |
|---:|:---:|:---:|
| datasets | FCN | TCN |
| GunPointAgeSpan | 98.73 | 97.15 |
| FordA | 89.77 | 91.67 |
| FordB | 79.01 | 80.37 |
| MelbournePedestrian | 90.13 | 95.17 |
| NATOPS | 97.22 | 95.56 |
| ElectricDevices | 69.51 | 69.47 |

Table 1: The test accuracy in % of the classification task.

#### B.3.1 FULLY CONVOLUTION NETWORK

We use a slightly modified FCN, similar to Wang et al. (2017). Ours contains four convolution blocks with a convolutional layer, a batch normalization layer (Ioffe & Szegedy, 2015) and a ReLU layer in each block. The kernel shapes and numbers of filter for convolution layers are $\{7, 5, 3, 3\}$ and $\{16, 32, 32, 16\}$. Therefore, there are four convolutional layers. Each convolutional layer has unit stride and no padding, which means the time sequence will be reduced continuously by the blocks. The final convolution block, which is behind the four convolution blocks, does not have a ReLU layer and contains a 1x1 convolutional layer. The 1x1 convolutional layer serves as a projection layer. It can not only reduce the channel size of feature maps but also keep their salient features. Finally, we apply Global Max Pooling on the features maps, before the softmax operation.

#### B.3.2 TEMPORAL CONVOLUTION NETWORK

For the TCN, that was first proposed by (Bai et al., 2018), we use a global pooling layer for the prediction. In our architecture, TCN has the convolution filters $\{16, 32, 32, 32\}$ and kernel shapes for convolution layers $\{7, 5, 5, 5\}$ in four residual blocks. Therefore, the total number of layers is $8$.

#### B.3.3 BIDIRECTIONAL LONG SHORT-TERM MEMORY

We use a standard, single-layer bi-LSTM to predict dense labels for the segmentation task, as implemented in PyTorch, with 512 hidden units. There is a dense layer behind the LSTM model to fit the hidden units 512 to the number of classes. Also, the dropout rate is set to 0.2 to prevent overfitting.

### B.3.4 U-TIME

We use the U-time (Perslev et al., 2019) architecture with the following configuration. In each convolution block, there are two dilated convolution layers with dilation 3, followed by a ReLU layer, a batch normalization layer, and a Max-Pooling layer at the end. The number of filters for convolution layers in four convolution blocks are $\{16, 32, 64, 128\}$ and the pooling window sizes are all 2. Two additional convolutions with filter numbers $\{256, 256\}$ follow after four convolution blocks. In each transposed convolution block, a nearest-neighbor up-sampling (Odena et al., 2016) of its input is implemented, followed by a dilated convolution layer with dilation 3, a ReLU layer and a batch normalization layer. The number of filters for convolution layers in four transposed convolution blocks are in the reverse order of the encoder $\{128, 64, 32, 16\}$. The kernel size of convolution layers in both encoder and decoder is 7.

## C ADDITIONAL RESULTS

We report additional results to supplement our discussion for a bi-LSTM architecture on the segmentation task in Sec. C.1, and for the classification task in Sec. C.2.

### C.1 EVALUATION ON SEGMENTATION TASK: BI-LSTM

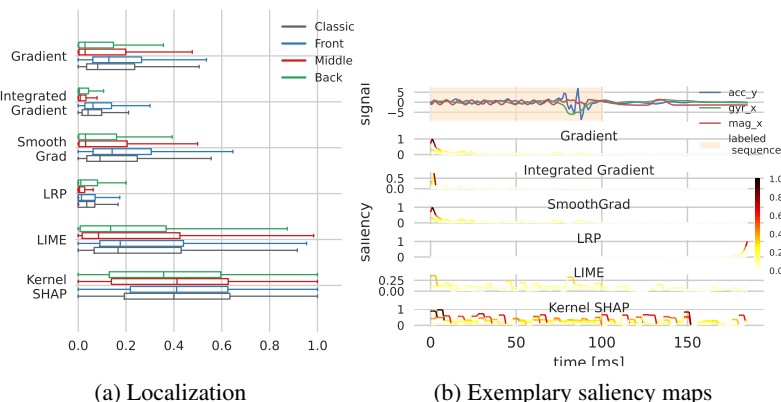

(a) Localization     (b) Exemplary saliency maps

Figure 6: We show the localization metric results for all methods on the tool tracking dataset and with the bi-LSTM model. (a) shows the results for the localization metric. (b) shows exemplary saliency maps.

The results for bi-LSTM on the segmentation task for tool tracking in Fig. 6 show that none of the visual interpretation methods, that are compatible with the model's architecture, produce satisfactory saliency maps.

### C.2 CLASSIFICATION TASK

We show the variability of each visualization method with respect to each dataset and metric category for the TCN in Fig. 7a. It is noteworthy that some datasets are more problematic for the methods, especially for faithfulness and intra-class stability, than others. We also show that the model has only a small influence. For this result, we aggregate all datasets and visualization methods separately for FCN and TCN in Fig. 7c.

In order to compare methods independent of datasets, we control for their bias by normalizing the scores of all methods for each {category, dataset} so that their mean is 0 and variance is 1. With this, we can correctly assess the "relative" performance of the visual interpretation methods across different datasets. The plot in Fig. 7b shows these relative (or marginal) scores that a method can achieve, compared to other methods on the same datasets.

We prove the orthogonality of metrics in Figure 8. No combination of our metric scores shows a high correlation. This shows that each metric measures a quality independent of all other metrics.

We show examples for saliency of the TCN architecture on the FordB dataset for the visual interpretation methods for class 0 in Figure 9 and class 1 in Figure 10. We argue that the choice of a suitable method to generate saliency maps should be guided by our framework.

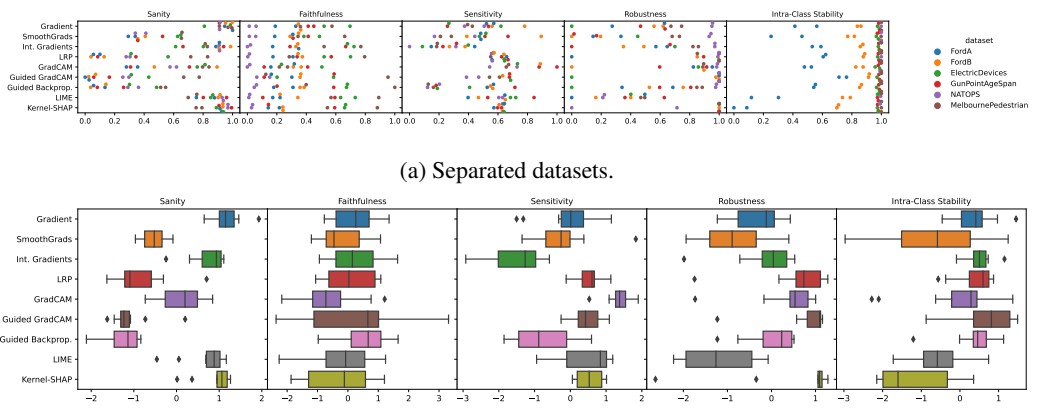

(a) Separated datasets.

(b) Relative scores that a method can achieve compared to other methods on the same datasets.

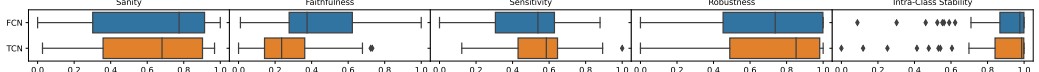

(c) Results for different model architectures, aggregated across datasets and visual interpretability methods. Model architecture has a comparatively small influence on the performance, which speaks to their generalization capabilities.

Figure 7: (a) shows scores separately over all {model, method } combinations. (b) removes the datasets' bias in order to assess the performance of individual visual interpretability metrics independent from datasets. (c) shows results for different model architectures, aggregated across datasets and visual interpretability methods. Model architecture has a comparatively small influence on the performance, which speaks to their generalization capabilities.

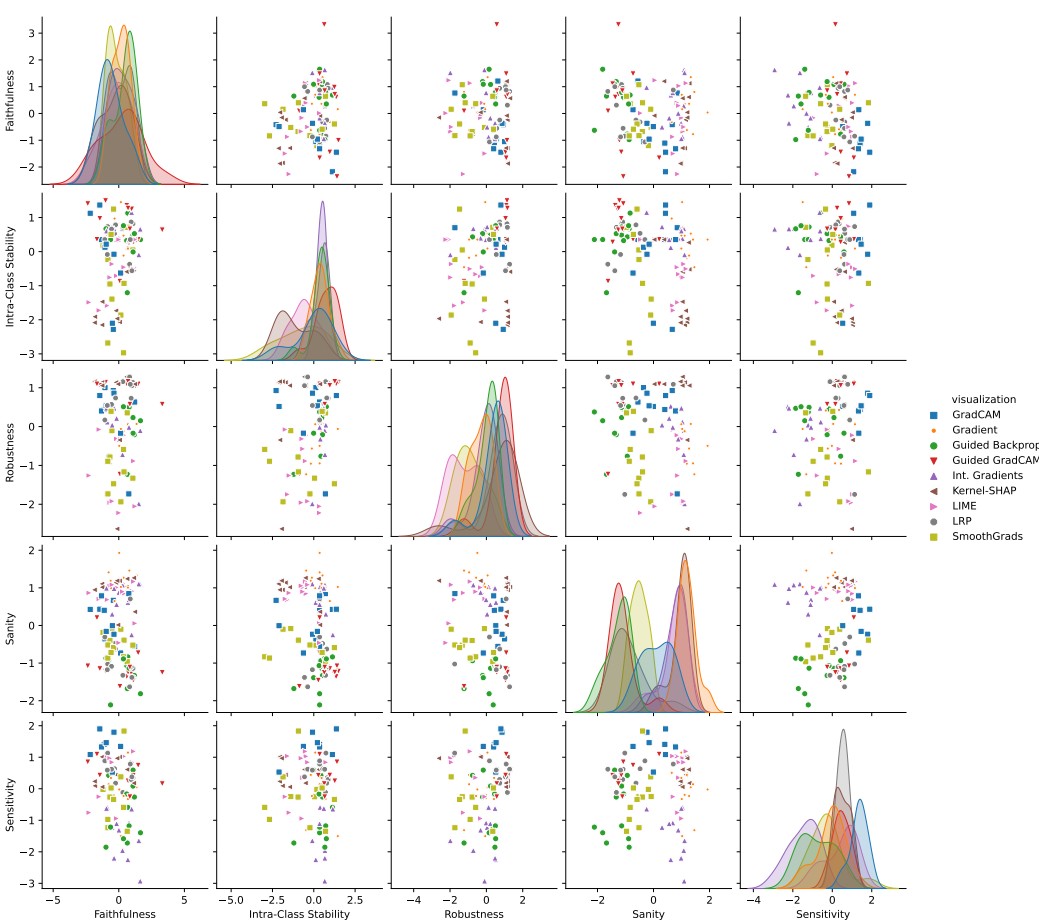

Figure 8: Pairplots of the correlations of metric scores with each other. No combination of metrics has a meaningful correlation with each other, proving that they provide independently useful signals. This analysis is based on scores that were normalized for dataset bias, but the results also hold when this normalization is not done.

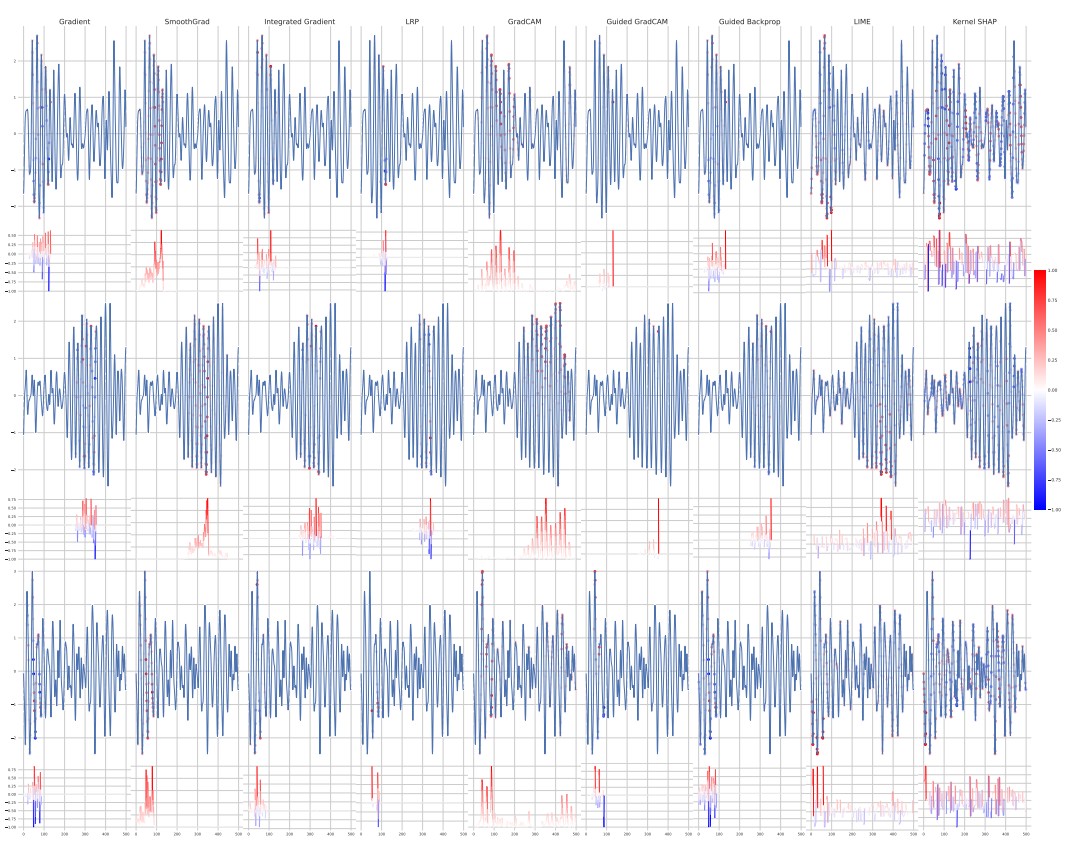

Figure 9: Examples of saliency maps for the TCN architecture on the FordB dataset for class 0.

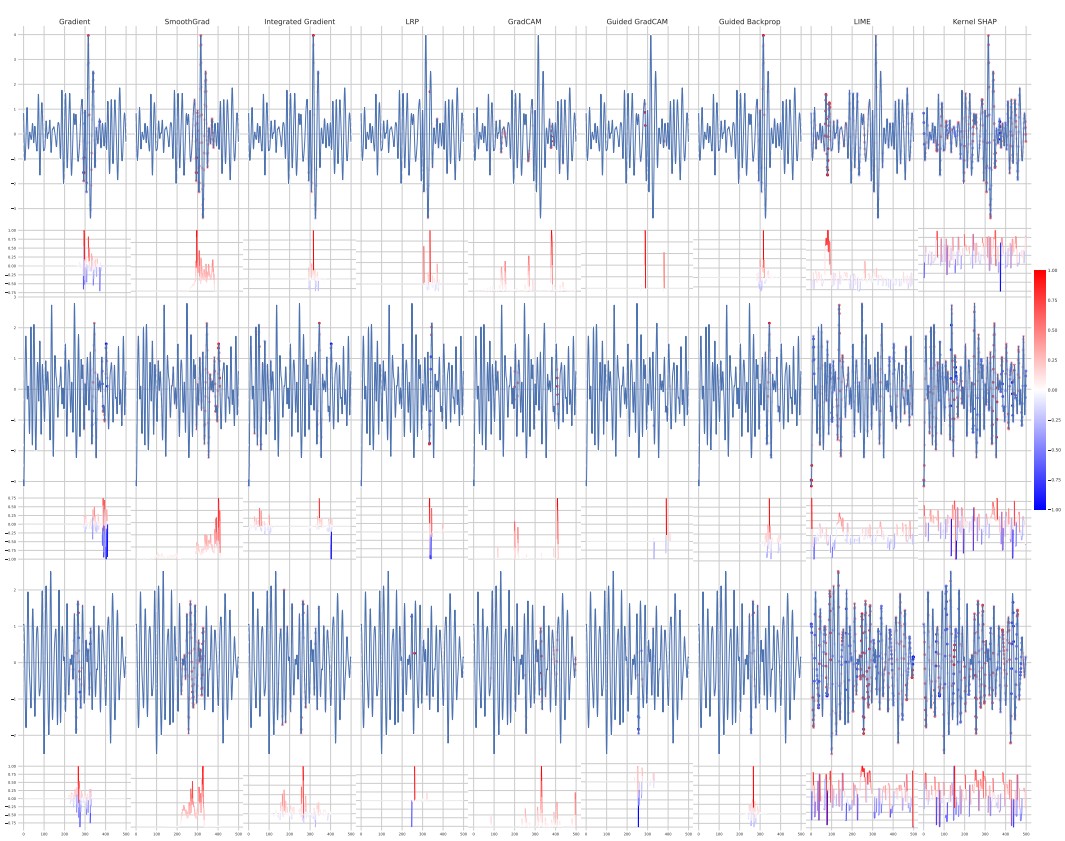

Figure 10: Examples of saliency maps for the TCN architecture on the FordB dataset for class 1.

