# OpenReview forum: "Objective Evaluation of Deep Visual Interpretations on Time Series Data"
_ICLR.cc/2022/Conference — ICLR 2022 Submitted_

### Official Review · Reviewer_9Qic · 2021-11-01

**Correctness:** 2
**Technical Novelty And Significance:** 1
**Empirical Novelty And Significance:** 1
**Recommendation:** 3
**Confidence:** 4

**Main Review:**

1) **Unclear objective / no novelty or insight**: The objective of this paper is very unclear, and there does not seem to be any novelty in the methods it introduces. Majority of the saliency evaluation methods it uses are known in literature, and these evaluation methods do not seem to change for time series classification. The paper simply proposes to use these metrics to evaluate saliency methods. Now, it is understandable that different saliency metrics are not aligned, but the paper proposes no solution for this. If each saliency method has different strengths and weakness, how can one use this "framework" to guide selection of saliency methods? This lack of a solid objective within this paper makes it unsuitable for publication at this time.
2) **Incorrect to use robustness of saliency map as a metric**: As a general remark on saliency metrics, please note that the only job of an evaluation metric should be to judge how well a saliency method reflects model behaviour. Metrics such as robustness or intra-class stability are thus contingent on such model behaviour. For example, if the underlying model is non-robust, there is no reason to expect robustness from saliency methods, and if a saliency method is indeed robust in such scenarios it does not reflect model behaviour and thus can be misleading. Thus usage of these metrics for evaluating saliency maps must be avoided.
3) **Incorrect to use localization of saliency map as a metric**: A similar critique holds for the localisation metric (including the pointing game and other metrics in literature). The implicit assumption here is that model holds the same parts of the input to be important as judged by a human evaluator. However models can be erroneous and can sometimes have spurious correlations and might assign a different portion of the input to be important. In such a case, we would want the saliency map to reflect modelling errors, which would necessarily perform poorly in the localisation metric. Hence localization is a not a good saliency evaluation metric.

**Summary Of The Paper:**

This paper proposes to use six evaluation metrics to evaluate saliency methods for time series classification. The experiments perform this evaluation for segmentation tasks, and find that few metrics achieve high scores with low variance.

**Summary Of The Review:**

The paper does not present either a novel method or any novel insight. For this reason, I would recommend rejection.

The post-rebuttal discussions did not change my view of the paper, but I am improving my score in light of author updates.

---

> ### Author Response · Authors · 2021-11-09
> **Authors' response**
>
> Thank you for your critical feedback! You raised some important points that we'd like to consider in our paper as well as in our response below.
>
> While we also see many points raised in the review, we do not fully agree that the objective of the paper is unclear (however, we of course take your concerns into account and make it even clearer). We provide an objective tool ("framework") that allows a user to select the most suitable saliency metric for a given {model, dataset}. We evaluate methods on the classification and segmentation task. To the best of our knowledge, we are the first to evaluate and prove that our chosen metrics are indeed orthogonal on time-series data. Furthermore, we provide guidance on the selection of saliency scores from a practitioner's standpoint. We consciously did not propose a new "final" aggregated score (based on metrics or a different evaluation), as any single aggregation will have severe problems. Instead, we provide a focused look on important metrics, and give instructions (at the end of Sec. 4.3) on how to interpret and use these results. To improve the paper: should we highlight or change that paragraph to make the objective clearer?
>
> **Model behavior/Robustness:** We agree that saliency metrics should essentially evaluate how well saliency captures model behavior. In accordance with your criticism of the score of adversarial robustness, we propose to tweak our score by factoring in the model's own robustness. We measure the model behavior under adversarial attacks and can use this to relativize the robustness score of the saliency methods. By explicitly accounting for the model robustness, we can provide a score that only measures if each interpretability method accurately captures the model's interpretability. Would this be sufficient as an additional novel insight?
>
> **Model behavior/Stability:**
> Intra-class stability is highly dependent on the dataset. While it is similar to a localization metric on a classification dataset, its features' location in the inputs are not know. Therefore, we are not able to factor the model's predictive errors out from the stability metrics. Hence, future work should normalize the scores via ground truth-based (or faithfulness-based) feature localization.
>
> **Localization metric:** We do not share the general critique of annotation-based saliency metrics. These metrics have a place in the literature on segmentation for evaluating saliency maps. Furthermore, we explicitly state the assumption in the paper, that localization is based on human annotators. Therefore, the saliency metric tests for exactly this.
>
> We do agree regarding your assessment, that the predictive performance of the underlying model on the segmentation task must be accounted for by researchers. Localization scores for the saliency are higher if the model performs better in the segmentation task. And they are lower when the model performance is low. Another version of the localization score would thus factor in the model's performance on the segmentation task and measure the saliency quality with respect to the input features' importance, e.g., by computing point-wise faithfulness scores. Our proposed method measures the saliency's quality with respect to the annotated sequence.
>
> However, we still see the value of our contribution to the field of segmentation because our method is rooted in annotator-based saliency metrics that are easily understandable and based in ground-truth. Such methods are feasible, they are widely used, and since they are typically not used in isolation from other analysis, the downsides that you commented on are compensated for.

---

> > ### Comment · Reviewer_9Qic · 2021-11-10
> > **Objective still unclear**
> >
> > > We provide an objective tool ("framework") that allows a user to select the most suitable saliency metric for a given {model, dataset}.
> >
> > The paper evaluates saliency methods on a set of 6 previously known metrics. I do not see how this constitutes a framework to guide selection of a suitable saliency metric for a given {model, dataset}. I re-read section 4.3 as you mentioned, but I fail to see any concrete recommendations for saliency map choice. Is there something I am missing?
> >
> > > We measure the model behavior under adversarial attacks and can use this to relativize the robustness score of the saliency methods.
> >
> > Note that this does not fix the problem. Adversarial attacks construct examples where the input changes slightly, yet the model output changes by a large amount. However we are interested in robustness of saliency maps or gradient vectors, where we require to construct examples where input changes slightly, yet the gradient vector changes by a large amount. Thus the two are not identical, and in fact it is possible to change gradient vectors without even changing model outputs. There is a large literature on performing such adversarial attacks. For example, see, "Explanations can be manipulated and geometry is to blame", Dombrowski et al, NeurIPS 2019. (https://arxiv.org/abs/1906.07983)
> >
> > > We do agree regarding your assessment, that the predictive performance of the underlying model on the segmentation task must be accounted for by researchers
> >
> > It is not just about model performance, and it is about detecting highly performant models that look at the wrong parts of the input for classification. In such cases we would like the saliency map to be faithful to model behaviour, and *not* place attributions on the right parts of the input as prescribed by localization metrics. It is possible for highly performant models to look at the wrong parts of the inputs and this is called the spurious correlations problem, or the shortcut learning problem (https://arxiv.org/abs/2004.07780). Thus localization metrics can be misleading and should not be used.
> >
> > E.g.: Consider the task of classifying cows vs camels. However let's say an image classification model simply detects the presence of background, i.e., grass or sand, to classify cows or camels respectively. In such a case a good saliency method should look at the background, i.e., grass or sand as that is what the model looks at. However metrics like the pointing game penalize such metrics, which is undesirable. This cow vs camel example is present in the following paper: https://arxiv.org/abs/1907.02893 .
> >
> > I would be happy to reconsider my decision if I have indeed missed something important regarding the paper. However so far I do not see any new insights or concrete recommendations made by the paper regarding selection of saliency methods.

---

> > > ### Author Response · Authors · 2021-11-10
> > > **Authors' response**
> > >
> > > > The paper evaluates saliency methods on a set of 6 previously known metrics. I do not see how this constitutes a framework to guide selection of a suitable saliency metric for a given {model, dataset}. I re-read section 4.3 as you mentioned, but I fail to see any concrete recommendations for saliency map choice. Is there something I am missing?
> > >
> > > Thank you for your further clarification, you are right. Indeed, while the steps to choose a saliency map seem very clear to us, they are apparently not stated as clearly in the paper itself. Hence, we propose to distill and extend the recommendations from the current draft into a more concrete framework and give it more room. To this end, we plan to add a dedicated section with clear guidance and reference the relevant experimental results as support. Would that meet your expectations?
> > >
> > > > Note that this does not fix the problem. Adversarial attacks construct examples where the input changes slightly, yet the model output changes by a large amount. However we are interested in robustness of saliency maps or gradient vectors, where we require to construct examples where input changes slightly, yet the gradient vector changes by a large amount. Thus the two are not identical, and in fact it is possible to change gradient vectors without even changing model outputs.
> > >
> > > Thank you for your comment (and for the appealing discussion!). We value your critique on this, and thank you for your pointers to adversarial attacks. We agree that evaluation metrics for visual interpretability methods (VIMs) need to quantify how much the VIM reflects model behavior.
> > > We measure a direct notion of this with faithfulness: "Does changing model inputs based on the saliency change the model output?".
> > > However, we fully agree that saliency maps also should be robust against adversarial attacks that *do not actually change how the model operates*.
> > > The aim of our robustness metric is to quantify how explanations change based on input perturbations - so, in other words, how susceptible they would be to adversarial attacks similar to the ones in Dombrowski et al.
> > > In order to measure *orthogonal* metrics, we chose to use a metric that does not consider the faithfulness of the VIM on perturbated samples as well (i.e., "how well the VIM reflects model behavior"). Instead, we quantify changes to saliency maps based on input perturbations directly.
> > >
> > > This metric is only valid (i.e., the score is valid and important to consider when choosing VIMs) under two conditions:
> > > - The VIM is faithful; i.e., it accurately reflects model function. As we noted, we measure this directly in our faithfulness metric.
> > > - The model is robust against adversarial perturbations. This condition, as you noted, is important to ascertain that we are not just measuring the robustness of the VIM on a model that is not robust itself.
> > >
> > > If these conditions hold, our robustness metric quantifies how robust the VIM is against adversarial attacks on the VIM itself, which is an important consideration when choosing a VIM.
> > > We are also currently running experiments that allow us to directly quantify how robust model outputs are in our dataset. This would allow us to show where condition 2 holds, and thus prove where our robustness metric is valid. We will add these conditions for robustness and our new section for recommendations, and add our results on the validity to the appendix.
> > >
> > > > In such cases we would like the saliency map to be faithful to model behaviour, and not place attributions on the right parts of the input as prescribed by localization metrics.
> > >
> > > Thank you for clarifying your point, and for providing the relevant references.
> > >
> > > We recognize that our paper lacks a discussion on this property of the localization metric (and similar metrics like Pointing Game). We will include this in the section on recommendations more explicitly. In that section, we propose that an ordering of metrics that should be followed. As you mentioned above, this was not yet stated clearly enough. We briefly give faithfulness a higher priority in our framework at the end of Sec. 4.3, however, this should become clearer.
> > >
> > > Based on the recommendations, we would still maintain that all categories are important. As you point out, the faithfulness metric measures whether saliency reflects model behavior, which is, among other things, important for discovering the shortcut learning problem. If we know a saliency is faithful, then the localization metric measures semantic precision of the saliency maps (i.e., annotation-based).
> > >
> > > Hence, we propose clearer recommendations for choosing saliency maps, paired with emphasis on the priority/ordering with the appropriate reasoning (including the spurious correlation problem; pointing out pitfalls). Would this contextualization of the localization metric meet your expectations?
> > >
> > > **Update on Nov. 12th 2021**
> > >
> > > Did you get a chance to read our reply? We have updated the paper. Does this meet your requirements?

---

> > > > ### Comment · Reviewer_9Qic · 2021-11-21
> > > > **Thank you for updates**
> > > >
> > > > Thank you for updating the paper and providing clear recommendations regarding selection of saliency maps. I will increase my score accordingly. Unfortunately, I still cannot recommend acceptance as the paper still does not have any novelty or insight.

---

### Official Review · Reviewer_oySW · 2021-11-02

**Correctness:** 4
**Technical Novelty And Significance:** 2
**Empirical Novelty And Significance:** 3
**Recommendation:** 6
**Confidence:** 2

**Main Review:**

This paper studies deep neural network (DNN) explainability methods in the context of time-series data. Several metrics exist for evaluating the validity of DNN explainability methods on computer vision tasks. However, it is not clear whether these metrics are reliable for DNNs on time-series data. This paper conducts experiments to compare 6 different metrics on time-series classification and segmentation tasks.


Strengths:
1. The paper is generally well organized. (ALTHOUGH there are minor typos scattered all over so I urge the authors to proof read their work).
2. The work is really well motivated. It is important to validate explainability methods before deploying them, and to that end it is important to design good validation criteria.
3. I found the experiments and the accompanying figures to be really well designed and insightful, thank you! In particular I liked figures 1, 4, and 5. The appendix also has insightful experiments and figures!

Weaknesses:
1. Can you explain how Fig. 2 is obtained? I think the observation in this figure is interesting in its own right. It is worrying that 6 different methods can have very low correlations.
2. I think the discussion in section 4.3 is hard to read and not very well organized. But I also understand that there are a lot of variables (data, task, model, explainability method, and metrics) which makes it difficult to fully cover all of them in text. (I am not penalizing the paper for this).

General questions (no need to run more experiments just wondering):
1. Some metrics are exhibit high variability on particular datasets (e.g. Ford). I wonder what you think would happen if you tried transfer learning. For instance pretrain on a dataset where all metrics show relatively low variability, then finetune on FordA dataset?

**Summary Of The Paper:**

This paper studies deep neural network (DNN) explainability methods in the context of time-series data. Several metrics exist for evaluating the validity of DNN explainability methods on computer vision tasks. However, it is not clear whether these metrics are reliable when applied to DNN explainability methods on time-series data. This paper conducts experiments to compare 6 of those metrics on time-series classification and segmentation tasks.

**Summary Of The Review:**

The paper is well motivated, the experiments are well designed, and the insights are helpful to the community.

---

> ### Author Response · Authors · 2021-11-09
> **Authors' response**
>
> Thank you for your concise summary of the paper’s goal. We especially appreciate the positive feedback for the motivation and figures.
>
> **Regarding Fig. 2:** We obtained it by measuring the pairwise Pearson correlation of the metric scores between all five perspectives, over all model architectures and datasets. We removed the dataset bias as in Fig. 7b, hence making the scores independent from the datasets. As you have recognized, the result proves that there is a low correlation between all methods. However, we think this is actually acceptable, and preferable to the case where all metrics correlate with each other, since it allows us to judge different characteristics of the saliency maps separately and in a disentangled way. We will clarify this in an updated version of the paper.
>
> **Regarding Transfer Learning:** The idea to use transfer learning to achieve better scores for particular datasets (e.g., Ford) seems promising. For a metric like Faithfulness (which depends on model accuracy) this may have a measurable effect. However, we observed that some properties are inherent to the data (like whether the time-series are centered or not) and the different dimensionality/application domain makes transfer learning generally less applicable when working with time-series (in contrast to image processing).
>
> Thank you also for pointing out the need for proof reading. The typos escaped us, we will take care of them.
>
> **Update on Nov. 12th 2021**
>
> Did you get a chance to read our reply? In the meantime, we have updated the paper after proof reading.

---

> > ### Comment · Reviewer_oySW · 2021-11-22
> > **Response to Rebuttal**
> >
> > Thank you for the responses and revisions. I decided to keep my score the same.

---

### Official Review · Reviewer_cLNa · 2021-11-03

**Correctness:** 4
**Technical Novelty And Significance:** 2
**Empirical Novelty And Significance:** 2
**Recommendation:** 5
**Confidence:** 2

**Main Review:**

The strength of this paper is that it provides a comprehensive evaluation of nine well-known interpretation methods for NNs with multiple architectures using multiple data sets. It is considered to be a very good survey paper to understand the characteristics of each method. The technical contribution of this study is mainly in the evaluation metric called localization, which is described in 3.6. This metric itself is based on a reasonable idea as an interpretive method. On the other hand, most of the considerations are derived from experimental results, which is a bit insufficient in terms of proposing new ideas based on deep technical insights.

**Summary Of The Paper:**

This study deals with the issue of interpretability in the analysis of time series data using deep learning models. Six metrics for evaluating interpretation methods are introduced and nine existing interpretation methods are evaluated. One of these metrics is proposed by the author. One of the metrics is based on an experimental evaluation, which shows that none of the methods is superior in all the metrics.

**Summary Of The Review:**

The paper is excellent as a survey paper focusing on performance evaluation, but somewhat inadequate as a research paper pursuing technical originality.

---

> ### Author Response · Authors · 2021-11-09
> **Authors' response**
>
> Thank you for your valuable comments!
>
> We want to address your main criticism concerning the localization metric in an updated version of this paper. Our suggestion is to extend Sec. 3.6 with more theoretical, generalized considerations, from which we have originally derived the metric. This would extend Sec. 3.6 in addition to Appendix A, that already provides further considerations from Tatbul et al. (2018), that are also valid for our localization metric for saliency. Would this meet your expectations, or did we misunderstand something?
>
> **Update on Nov. 12th 2021**
>
> Did you get a chance to read our reply? In the meantime, we have updated the paper and included the changes. Does this meet your requirements?

---

> > ### Comment · Reviewer_cLNa · 2021-11-19
> > **Thanks for revising the paper.**
> >
> > Thanks for revising the paper. The modification makes the contribution clearer, while the main emphasis of the entire paper is still focusing on a comprehensive evaluation of well-known interpretation methods, and the technical contribution seems to be still weak, so I would like to remain the evaluation score as it is.

---

### Official Review · Reviewer_aJfy · 2021-11-08

**Correctness:** 3
**Technical Novelty And Significance:** 2
**Empirical Novelty And Significance:** 3
**Recommendation:** 6
**Confidence:** 4

**Main Review:**

Strengths:
- The paper is well written overall
- Experiments are well designed where the six metrics can be compared for different tasks
- The paper demonstrates why it is necessary to evaluate all six metrics for every visual interpretation
- The paper compares 9 different interpretation methods
- The paper shows how these metrics can vary across datasets and for different models

Weaknesses:
- The metrics are derived from existing literature
- Figure 5 caption needs to be fixed


**Summary Of The Paper:**

The paper proposes six quantitative metrics for evaluating post-hoc visual interpretation methods on time series. It demonstrates the efficacy of nine visual interpretation methods using these six metrics over some common neural network architectures.

**Summary Of The Review:**

The paper combines multiple interpretation approaches and evaluation metrics for those interpretation schemes. Thus, it demonstrates the need for multiple interpretation evaluation schemes to truly understand which ones are optimum. It also shows how these can vary across different datasets and models. So, although the paper is not novel, the experiments performed and the results obtained is very useful for researches looking for a way to perform post-hoc visualization. As a result, I consider it as a borderline paper.

---

> ### Author Response · Authors · 2021-11-09
> **Authors' response**
>
> Thank you for your valuable feedback!
>
> As you summarized, our aim is to provide a broad and useful set of metrics for saliency methods to researchers.
>
> Indeed, we derive several metrics from literature and adapt them to time-series. We acknowledge that we take existing metrics out of their original context and apply them to time-series (evaluate and benchmark). Additionally, we argue that the contribution of the localization metric (with its bias for front, middle or back) extends the state of the art in the literature, in particular for time-series segmentation.
>
> Thank you for pointing out the issue with the caption of Fig. 5. We will fix this in an updated version of the manuscript.
>
> **Update on Nov. 12th 2021**: Did you get a chance to read our reply? In the meantime, we have updated the paper and included the changes. Does this meet your requirements?

---

> > ### Comment · Reviewer_aJfy · 2021-11-22
> > **Thanks for the revision**
> >
> > Thanks for the revision. The technical contributions and novelty of the paper still appears to be weak, so I would like the evaluation score to remain as it is.

---

### Author Response · Authors · 2021-11-12
**Updated paper**

We want to thank the reviewers for their valuable feedback and discussions. In addition, we provide an updated version of the paper that addresses the comments.

The new version clarifies how Fig. 2 got obtained. We expanded the derivation of the Localization metric in Section 3.6 significantly. We also reorganized the discussion in Section 4.3. The new Section 5 elaborates on the recommendations that our framework makes in more detail. Finally, we improved form and typography.

We hope that the updated revision meets the requirements and answers open questions. We are available for discussions for the second half of the rebuttal period.

---

### Decision · Program_Chairs · 2022-01-20

**Decision:**

Reject

**Comment:**

This paper evaluates interpretation methods of neural networks on time series data. The reviewers find some values in this work, but were also consistently concerned with the main theme and novelty of this work. The authors have actively responded to reviewer comments, but the reviewers were not convinced with the major contributions and novelty. Thus the work is not ready for ICLR.